# Fabrication of Flexible Supercapacitor Electrode Materials by Chemical Oxidation of Iron-Based Amorphous Ribbons

**DOI:** 10.3390/ma16072820

**Published:** 2023-04-01

**Authors:** Mircea Nicolaescu, Melinda Vajda, Carmen Lazau, Corina Orha, Cornelia Bandas, Viorel-Aurel Serban, Cosmin Codrean

**Affiliations:** 1Department of Materials and Manufacturing Engineering, Faculty of Mechanical Engineering, Politehnica University Timisoara, Mihai Viteazu 1, 300222 Timisoara, Romania; mircea.nicolaescu@student.upt.ro (M.N.); viorel.serban@upt.ro (V.-A.S.); 2National Institute for Research and Development in Electrochemistry and Condensed Matter, Dr. A. P. Podeanu 144, 300569 Timisoara, Romania; melinda.vajda@student.upt.ro (M.V.); carmenlazau@yahoo.com (C.L.); orha.corina@gmail.com (C.O.); cornelia.bandas@gmail.com (C.B.); 3Department of Applied Chemistry and Engineering of Inorganic Compounds and Environment, Faculty of Industrial Chemistry and Environmental Engineering, Politehnica University Timisoara, Piata Victoriei 2, 300006 Timisoara, Romania

**Keywords:** amorphous ribbons, oxidation, supercapacitors, nanocrystalline Fe_2_O_3_

## Abstract

A flexible electrode constructed from Fe-based amorphous ribbons decorated with nanostructured iron oxides, representing the novelty of this research, was successfully achieved in one-step via a chemical oxidation method, using a low concentration of NaOH solution. The growth of metal oxides on a conductive substrate, which forms some metal/oxide structure, has been demonstrated to be an efficient method for increasing the charge transfer efficiency. Through the control and variation of synthetic parameters, different structures and morphologies of iron oxide were obtained, including hexagonal structures with a hollow ball shape and rhombohedral structures with rhombus-like shapes. Structural and morphological characterization methods such as X-ray diffraction and SEM morphology were used on the as-synthesized composite materials. The supercapacitor properties of the as-developed amorphous ribbons decorated with Fe_2_O_3_ nanoparticles were investigated by cyclic voltammetry, galvanostatic charge discharge, and electrochemical impedance spectroscopy. The flexible supercapacitor negative electrode demonstrates a specific capacitance of 5.96 F g^−1^ for the 0.2 M NaOH treated sample and 8.94 Fg^−1^ for the 0.4 M NaOH treated sample. The 0.2 M treated negative electrodes deliver 0.48 Wh/kg at a power density of 20.11 W/kg, and the 0.4 M treated electrode delivers 0.61 Wh/kg at a power density of 20.85 W/kg. The above results show that these flexible electrodes are adequate for integration in supercapacitor devices, for example, as negative electrodes.

## 1. Introduction

Over the past decade, due to the increase in energy demands, electrochemical supercapacitors have gradually become one of the most promising energy storage devices because of their low cost, long life, and high power density and capacitance characteristics. Supercapacitors have many practical uses that demand high power outputs and fast charge/discharge rates, such as electric vehicles, renewable energy systems, and consumer electronics. In addition, they are considered more environmentally safe compared to traditional batteries because they lack toxic chemicals and heavy metals [1]. The performance of supercapacitors in general depends on the electrode materials; thus, new electrode materials have been designed to increase the performance of supercapacitors [2]. However, the main problem of supercapacitors is currently the issue of low energy density. This obstacle can be overcome by employing materials possessing a high theoretical capacitance. Electrode materials can be classified into three groups: carbon-based materials [3], conducting polymers [4], and transition metal oxides [5]. One of the most promising and most used materials for fabricating electrodes in supercapacitors are porous-carbon-based materials. They have a large surface area, a high porosity, and a good electrical conductivity, which makes them a promising electrode material for storing electric charge [3]. Many researchers have approached these carbon-based materials with the aim of enhancing energy storage properties through various methods. Yifan Wang et al. doped porous carbon electrodes with nitrogen using a one-step activation carbonization method. By achieving a high N doping amount of 3.35 at.%, they obtained supercapacitor electrodes with a specific capacitance of 250 F g^−1^ at 50 A g^−1^ [6]. Gaigai Duan et al. obtained a nitrogen-doped carbon/ZnO composite electrode material by impregnating flax fibers with Zn(NO_3_)_2_ and EDTA Na_2_Zn activators, followed by thermal treatment. The prepared electrode exhibited a specific capacitance of 292 F g^−1^ at a current density of 0.5 A g^−1^ [7]. Another class of electrode materials is represented by conducting polymers, which are a promising material for pseudo-capacitor devices due to their high electrical conductivity and redox properties. Among the conducting polymers, polypyrrole, polyaniline, and derivatives of polythiophene are the most extensively studied materials for use in pseudocapacitor devices. These conducting polymers have demonstrated promising results due to their ability to store charge via redox reactions and their high electrical conductivity, which is promising for use in electrodes for storing electric charge [4]. Jehan El Nady et al. developed a nanocomposite electrode for supercapacitor applications by using a one-step electrodeposition technique to deposit polypyrrole on a NiO substrate, forming a polypyrrole/NiO nanocomposite electrode. Compared to the pristine Ppy electrode, the supercapacitor performance of the Ppy/NiO nanocomposite electrode was significantly enhanced. The study revealed that the Ppy/NiO electrode deposited at 4 A/cm^−2^ demonstrated the highest specific capacitance of 679 F g^−1^ at 1 Ag^−1^ [8]. Azza Shokry et al. synthesized a supercapacitor electrode by combining polythiophene and single-walled carbon nanotubes (SWCNTs) in varying ratios to form nanocomposites. The maximum specific capacitance was achieved when the nanocomposite material contained 50% SWCNTs and had a specific capacitance of 245.8 F g^−1^ at a current density of 0.5 Ag^−1^ [9].

Nanostructures based on transition metal oxides and hydroxides [10], such as Fe_2_O_3_, Fe_3_O_4_, Co_3_O_4_, NiO, CuO, and FeOOH, are promising electrode materials for supercapacitors due to their high specific capacitance [11]. Suprimkumar D. Dhas et al. synthesized porous NiO nanoparticles for use in supercapacitor applications using a simple hydrothermal method. The supercapacitor metal oxide electrode demonstrated a specific capacitance of 116 F g^−1^ at 10 mAg^−1^ in KOH electrolyte and a maximum capacitance of 74 F g^−1^ at 10 mAg^−1^ in Na_2_SO_4_ electrolyte [12]. M. M. Momeni et al. conducted a study where they used chemical corrosion to produce CuO nanostructures on a pure copper plate. Corrosion was carried out in a mixture of 20 mL NaOH and 35 mM (NH_4_)_2_S_2_O_8_ aqueous solution at room temperature for varying durations. The specific capacitance of the supercapacitor electrode material was determined to be 158 F g^−1^ using cyclic voltammetry at a scan rate of 10 m s^−1^ [13]. Fe_2_O_3_ is considered a very promising candidate for electrode materials for supercapacitors and batteries because of its high theoretical specific capacitance, non-toxicity, and low cost [14]. Various methods have been used to synthesize Fe_2_O_3_ nanostructures, such as sol-gel, hydrothermal, electrodeposition, and vapor deposition methods [15]. Similar to other transition metal oxides, Fe_2_O_3_ usually suffers from poor conductivity and particle agglomeration, which reduces its performance, mainly at high rates. The formation of metal/oxide structures through the growth of metal oxides on conductive substrates has been shown t−o be an effective way to address the issue of conductibility by enhancing the efficiency of charge transfer [5]. Dealloying and corrosion processes permit the growth of uniform metallic oxide on the porous metallic core, with a better interface between the metal and oxide, and improve the uniformity distribution of the active materials by avoiding particle agglomeration [16,17]. Iron oxide, with a high theoretical capacity, was obtained through a dealloying/corrosion process by using materials both in powder form and in the form of a three-dimensional layer grown on the ribbon surface in the pure phase or in a mix of oxides [18]. The chemical or electrochemical oxidation synthesis of Fe_2_O_3_, compared to the other methods, is the simplest approach to obtain metal oxides on the surface of metal ribbons with a low production cost and high reproducibility. Baolong Sun et al. conducted a study on the synthesis of iron oxides by oxidizing low carbon steel ribbons in a 1 M NaOH aqueous solution for 6 h at 60 °C. The authors compared the areal capacitance of the obtained iron oxide decorated ribbons with that of the same ribbons coated with a layer of polypyrrole for use in supercapacitors. The results showed that the areal capacitance of the rust coated with polypyrrole was significantly higher than that of the uncovered rust. The coated rust electrode exhibited a capacitance of 2.202 F cm^−2^ compared to 0.878 F cm^−2^ for the uncovered electrode at 1 mA cm^−2^ [19]. A.A. Yadav et al. synthesized iron oxide negative electrodes using a chemical method for use in asymmetric supercapacitor devices. The α-Fe_2_O_3_ thin films obtained using a simple chemical bath deposition (CBD) method exhibited a nanosphere-like morphology. At a current density of 4 Ag^−1^, the specific capacitance value of the α-Fe_2_O_3_ electrode was found to be 2125 F g^−1^ [20]. Jianhui Zhu et al. employed an iron rust byproduct from the steel industry, which was previously deemed useless and unwanted, and subjected it to a straightforward hydrothermal treatment in an aqueous solution containing HNO_3_. The resulting product formed a sphere-shaped structure that was integrated into a negative electrode for a storage device. The iron rust electrode obtained through this process exhibited a maximum specific capacity of 269 mA h g^−1^ at 0.3 A g^−1^ [21].

An amorphous alloy was chosen for the fabrication of uniform nano/microstructures because of its disordered atomic scale structure, absence of defects, grain boundaries, secondary phases, element segregation, and heterogeneous structure [22]. These properties, combined with its super-elasticity, make it an ideal candidate for obtaining electrodes for supercapacitors [23]. Additionally, by controlling the morphology and microstructure of the grown Fe_2_O_3_ layer, the electrochemical performance can be improved. With different morphologies and microstructures, major changes in the electrochemical and storage properties of these devices have been reported. In this study, a chemical oxidation process on iron-based amorphous ribbons in an alkaline solution was used to grow different oxide nanocrystalline structures and Fe_2_O_3_ morphologies, such as hexagonal with a hollow ball structure and rhombohedral with a rhombus-like structure. Furthermore, the novelty of the research consists of the use of Fe_75_Si_12_B_10_Nb_3_ amorphous ribbons as a substrate for flexible electrodes constructed on Fe-based amorphous ribbons decorated with nanostructured iron oxides fabricated in a one-step chemical oxidation synthesis for supercapacitor applications.

## 2. Materials and Methods

### 2.1. Fabrication of Amorphous Ribbons

Quaternary alloys with a nominal composition of Fe_75_Si_12_B_10_Nb_3_ were synthesized by the arc melting process. To ensure accurate atomic percentages and the homogeneity of the raw materials, the melting process was repeated four times, using a mixture of pure elemental Fe, Nb metal, and Fe–B and Fe–Si ferroalloys under an argon atmosphere. The raw materials were prepared to match the nominal atomic percent of the Fe_7_5Si_12_B1_0_Nb_3_ alloy. The melt spinning method [24] with a copper roll speed of 37.7 m/s was used to prepare amorphous Fe–Si–B–Nb ribbons with a 25 µm thickness and a 15 mm width. We used optical emission spectroscopy to determine the atomic percentage of the prepared sample, and ensured that the alloy compositions represented the nominal atomic percent. 

### 2.2. Fabrication of Flexible Electrodes Decorated with Iron Oxide Nanoparticles

The corrosion process was carried out in a NaOH aqueous solution (10 mL) of 0.2 M and 0.4 M concentrations in free immersion conditions for 7 days [25]. The as-oxidized ribbons were removed and washed thoroughly with distilled water to clean any residual impurities. These steps are presented in the schematic diagram in Figure 1.

The use of a NaOH solution can facilitate the transformation of Fe to FeOOH and Fe_2_O_3_ by creating an alkaline environment that can promote the formation of dehydrated iron oxide [25,26]. The concentration, the volume of the container, the amount of solution, and the temperature of the NaOH solution can directly affect the kinetics and thermodynamics of these reactions. These reactions need a long period of time and a stable environment to obtain the desired structure and morphology of iron oxide.

The chemical reactions that occur between the iron-based amorphous ribbons and the NaOH solution take place according to the following reactions [27]:Fe + 2NaOH + 0.25O_2_ + 0.5H_2_O → FeOOH + 2Na^+^ + 2OH^−^(1)
FeOOH (goethite) → γ-FeOOH (lepidocrocite) + H_2_O(2)
γ-FeOOH (lepidocrocite) → Fe_2_O_3_ (hematite) + H_2_O(3)

Hexagonal Fe_2_O_3_ structures in the form of hollow spheres were obtained by immersing the iron-based amorphous ribbons in a solution of NaOH with a concentration of 0.2 M at a temperature of 20 °C. The immersion process was carried out for a period of 7 days to allow the formation of a nanocrystalline structure.

To obtain orthorhombic and rhombohedral Fe_2_O_3_ structures, the material was immersed in a NaOH solution with a concentration of 0.4 M at a temperature of 25 °C for 7 days. During this process, the NaOH solution reacted with the Fe_2_O_3_ material, leading to the formation of the desired structures with a rhombus-like morphological shape.

### 2.3. Materials and Characterization

To determine the crystalline structure of the decorated oxides, X-ray diffraction (XRD) was used. XRD was performed using a PANalytical X’Pert PRO MPD diffractometer equipped with a monochromator used to filter out the fluorescent radiation, with Cu-Kα radiation (CuKa1: 1.540598 Å and CuKa2: 1.544426 Å) in the range of 2theta from 20 to 80°. The morphology of surface oxides was examined by scanning electron microscopy (SEM). An FEI Inspect S model coupled with an energy dispersive X-ray analysis detector (EDX) was used to inspect the surface morphologies. The electrochemical performances of the electrodes were determined on an electrochemical workstation (Voltalab Potentiostat model PGZ 402) using a standard system consisting of three electrodes, a flexible electrode (size 0.5 × 0.5 mm), a Ag/AgCl electrode (sat. KCl), and a Pt wire served as the working electrode, reference electrode, and counter electrode, in a 0.5 M Na_2_SO_4_ solution. Electrochemical measurements included cyclic voltammetry (CV), galvanostatic charge–discharge (GCD), and electrochemical impedance spectroscopy (EIS).

## 3. Results and Discussion

### 3.1. Structural and Morphological Properties

Structural investigations of the as-prepared flexible electrode before and after being decorated with Fe_2_O_3_ nanoparticles were carried out using X-ray diffraction (XRD) analyses (Figure 2). In the XRD patterns of the as-spun ribbons, a broad intensity peak was observed at a 2θ angle within the range of 37 to 50°. The shape of the diffraction peak was in line with other research studying iron-based amorphous ribbons [28,29]. This peak is indicative of the amorphous state of the iron-based alloy, as it is also present in the spectra of metallic ribbons decorated with iron oxide. This furthermore evidenced the presence of an amorphous core that gave the flexible application of the supercapacitor electrode. In the case of the ribbon immersed in a 0.2 M NaOH solution, α-Fe_2_O_3_ with diffraction peaks at 7.8° (002), 25.2° (004), 30° (106), and 47.3° (027) associated with a hexagonal structure (JCPDS No 01-076-1821) is present. Additionally, a rhombohedral structure (JCPDS No 01-073-0603) of α-Fe_2_O_3_, with diffraction peaks at 21.2° (012), 35.5° (110), 40.4° (113), and 53.7° (211) is also present in the sample maintained in a 0.4 M alkaline solution.

Scanning electron microscopy (SEM) was used to study the morphology and uniformity of the surface oxide layer. In Figure 3a, the surface morphology of flexible ribbons decorated with oxide nanoparticles at an immersion time of 7 days and a concentration of NaOH solution of 0.2 M is presented. As a result of the high holding time of the ribbons in the alkaline solution, a porous hollow-ball-like oxide [30] is present on the surface of the flexible amorphous ribbons. The mesoporous structure of the oxide particles can increase the specific surface area of the film. The surface morphology of the oxide-decorated flexible ribbon treated in a solution of 0.4 M NaOH for 7 days is presented in Figure 3b, and it can be observed that uniform structures composed from rhomboidal formations are obtained. Figure 3c,d presents the cross-section images of the as-obtained electrodes, and it is observed that the oxide layer thickness is directly proportional to the concentration of the NaOH solution. Typically, hematite forms in a hexagonal crystal system with a trigonal structure. However, under certain conditions, hematite can exhibit a rhombohedral crystal structure, which gives it a rhombus-like appearance [31,32]. The optimization of the process parameters lead to the formation of a uniform oxide layer distributed on the surface of the samples.

### 3.2. Electrochemical Performance of the Flexible Electrodes

Cyclic voltammetry (CV) studies of the hexagonal and rhomboidal Fe_2_O_3_-decorated electrodes were undertaken to assess the electrochemical behavior of the electrodes, and are presented in Figure 4a,b. 

The measurements were performed at a potential window range of −0.7 to 0 V, using scan rates of 0.005, 0.01, 0.02, 0.05, and 0.1 V s^−1^, and both materials presented negative electrode behavior [15,20].

The recorded current increased with the increasing scan rate for both flexible electrodes decorated with Fe_2_O_3_ nanoparticles. This observation suggests that the electrodes exhibit excellent supercapacitive behavior. The shape of the CV profile provides valuable information about the mechanism of charge storage and the performance of the electrode. The non-rectangular shape of the cyclic voltammetry (CV) profile is linked to the occurrence of faradaic reactions, namely reduction and oxidation, on the electrodes [33].

The capacitance (C_P_) derived from the CV analysis was calculated according to Equation (4) [34]:(4)$$C_{P}=\frac{A}{\mathrm{km}\Delta V}$$
where C_P_ is the capacitance, A is the area under the curve, k is the scan rate, m is the total mass of the sample, and ΔV is the potential window.

The calculated capacitance versus the scan rate of the CV analysis for both negative electrodes is plotted in Figure 4c. At a 0.005V scan rate, the capacitance values are 16.25 F g^−1^ and 19.5 F g^−1^ for the amorphous ribbon treated with 0.2 M NaOH solution and 0.4 M alkaline solution, respectively. Figure 4c shows a clear decrease in the charge storage capacity from 16.25 to 4.74  F g^−1^ for the 0.2 M electrode and from 19.5 to 13.3  F g^−1^ for the 0.4 M electrode with increasing scan rate. This decrease is attributed to the ion exchange mechanism and is more pronounced for the 0.2 M electrode than for the 0.4 M electrode, probably because the highly porous hollow structures of the electrode need more time for the intercalation–deintercalation process during charge and discharge [35,36]. Consequently, only a small fraction of the electrode material can be utilized, whereas most of the material remains unutilized at high potentials.

Figure 5a,b illustrates the galvanostatic charge–discharge (GCD) curves as a function of time at different current densities (0.1, 0.2, 0.3, 0.4, and 0.5 A g^−1^) for the negative electrodes (0.2 M and 0.4 M) to assess the electrode’s performance. 

As previously reported in the literature for various electroactive materials, an increase in the current density results in a decrease in the discharging time [37,38]. The GCD profiles for the negative electrode present non-linear behavior (more visible at a low current density), indicating that the capacitance of the supercapacitor is not constant over the entire charging/discharging process [35,39]. Furthermore, this indicates that the Faradic pseudocapacitive nature of the as-produced flexible electrodes agrees with the results obtained from the cyclic voltammetry measurements.The GCD curves of the 0.2 M tested electrode displays two main variation regions (Figure 5a): an initial rapid voltage drop due to the internal resistance followed by a subsequent linear region that can be attributed to the capacitive behavior [40]. The GCD curve of the 0.4 M-treated electrode exhibits a third region, which is likely caused by Faradaic processes, such as redox reactions or electrochemical adsorption/desorption, occurring at the interface between the electrode and the electrolyte [35].

The specific capacitance (C_SP_), energy density (E), and power density (P) were calculated from the GCD analysis according to the following formulas [37]:(5)$$C_{\mathrm{SP}}=\frac{I\Delta t}{\Delta \mathrm{Vm}}$$
(6)$$E=\left(\frac{{\;C}_{\mathrm{SP}}{\Delta V}^{2}}{2}\right)$$
(7)$$P=\frac{\Delta \mathrm{VI}}{2m}$$
where I represents the applied current (in A), m represents the mass of the active material (in g), and Δt and ΔV represent the discharging time (in s) and the discharge voltage, respectively.

In Figure 5c, the specific capacitance, calculated from the GDC analysis, is plotted against the current density. After increasing the power density from 0.1 A g^−1^ to 0.5 A g^−1^, the specific capacitance of the 0.2 M electrode decreased from 5.96 F g^−1^ to 0.9 F g^−1^ and showed a more pronounced decrease compared to the 0.4 M sample. The 0.4 M sample exhibited a higher specific capacitance compared to the 0.2 M sample, and the capacitance was also found to be more stable, with only a slight decrease from 8.94 F g^−1^ to 7.41 F g^−1^ when the power density increased from 0.1 A g^−1^ to 0.5 A g^−1^. Figure 5d displays the specific energy vs. specific power calculated from the GDC analysis at different current densities.

The behavior of the two tested electrodes differs in terms of energy by the amount of stored energy and the rate at which that energy can be delivered, as shown in the graph. However, the specific energy to specific power ratio remains constant for both electrodes, decreasing with the increasing current density. Both electrodes deliver the highest energy density at an applied current density of 0.1 A g^−1^. The 0.2 M electrode delivers 0.48 Wh/kg at a power density of 20.11 W/kg and the 0.4 M electrode delivers 0.61 Wh/kg at a power density of 20.85 W/kg.

Electrochemical impedance spectroscopy (EIS) was performed over a frequency range of 0.1 Hz to 10,000 Hz with an amplitude of 0.01 V. The EIS spectra for 0.2 M and 0.4 M electrodes exhibit a semicircle and a line, as depicted in Figure 6a,b. The intersection of the EIS plots and the real axis (an impedance of zero) represents the ohmic resistance, which encompasses the ionic resistance of the electrolyte, the resistances of the iron oxide and iron base amorphous ribbons substrate, and the contact resistance, R_S_. The semicircle observed in the EIS analysis is an indication of the charge transfer resistance (R_P_) between the amorphous ribbons decorated with iron oxide and the electrolyte [41]. This resistance corresponds to the electrochemical activity of the active material in the system (the impact of both Faradaic and non-Faradaic reactions) [35]. The slope of the straight line observed in the low frequency region is attributed to the Warburg resistance and capacitive behavior of the electrode [42]. Both electrodes have a good capacitive behavior, demonstrated by an inclination greater than 45° with the decrease in frequency [35]. The 0.4 M electrode displays a higher capacitive behavior than the 0.2 M electrode, as evidenced by a steeper inclination in the frequency range near 90°. The inset of Figure 6a,b illustrates the equivalent circuit used for curve fitting. Based on the extracted parameters, it was observed that the 0.4 M negative electrode exhibited a lower ionic and electronic resistance, R_S_ = 5.44 Ω, compared to the 0.2 M negative electrode, which had R_S_ = 10.1 Ω. This suggests that the 0.4 M negative electrode has better conductivity. Moreover, the 0.4 M negative electrode has a lower interfacial charge transfer resistance, R_P_ = 3.30 Ω, compared to the 0.2 M negative electrode at R_P_ = 55 Ω. This can explain the superior performance of the 0.4 M negative electrode.

## 4. Conclusions

In this work, a flexible negative electrode for supercapacitors was successfully prepared via a one-step chemical oxidation process by decorating amorphous ribbons with Fe_2_O_3_ nanoparticles. This process presumes the immersion at a high holding time of the iron-based amorphous ribbons in a low molarity NaOH solution. Uniform mesoporous hollow-ball-like oxide (0.2 M) and rhombus-like oxide (0.4 M) formations are obtained on the alloy surface. Varying the process parameters, such as the concentration, temperature, or holding time, during the oxidation process led to different morphologies or crystal structures. These different structures can be further optimized for other specific applications. From the XRD data, it was found that the hollow ball morphology samples have a hexagonal structure, and the rhombus-like morphology samples have a rhomboidal structure. The CV curves show that both samples work like negative electrodes and show a charge storage capacities of 16.25 F g^−1^ for the 0.2 M sample and 19.5 F g^−1^ for the 0.4 M sample at a 0.05 V/s scan rate. From the GCD analysis, the maximum specific capacitance was obtained, with values of 5.96 F g^−1^ for the 0.2 M sample and 8.94 Fg^−1^ for the 0.4 M sample at a power density of 0.1 A g^−1^. Additionally, both electrodes deliver the highest energy density at an applied current density of 0.1 A g^−1^. The 0.2 M electrode delivers 0.48 Wh/kg at a power density of 20.11 W/kg and the 0.4 M electrode delivers 0.61 Wh/kg at a power density of 20.85 W/kg. In conclusion, this one-step synthesis process is a simple, low-cost, effective, and efficient method for large-scale production of flexible negative electrode materials. According to the obtained data, the amorphous ribbons decorated with Fe_2_O_3_ nanoparticles have high potential in supercapacitor applications. The relatively small values of current density can be explained by the weight of the electrode, the total weight of the amorphous metal alloy electrode, and the iron oxide being taken as the electrode mass.

## Figures and Tables

**Figure 1 materials-16-02820-f001:**
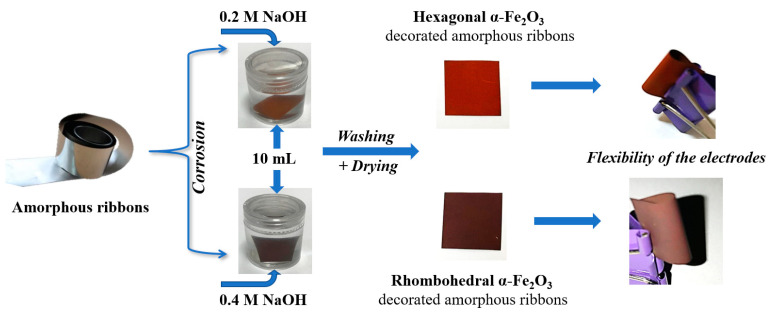
Schematic illustration of the one-step synthesis process of the flexible electrode.

**Figure 2 materials-16-02820-f002:**
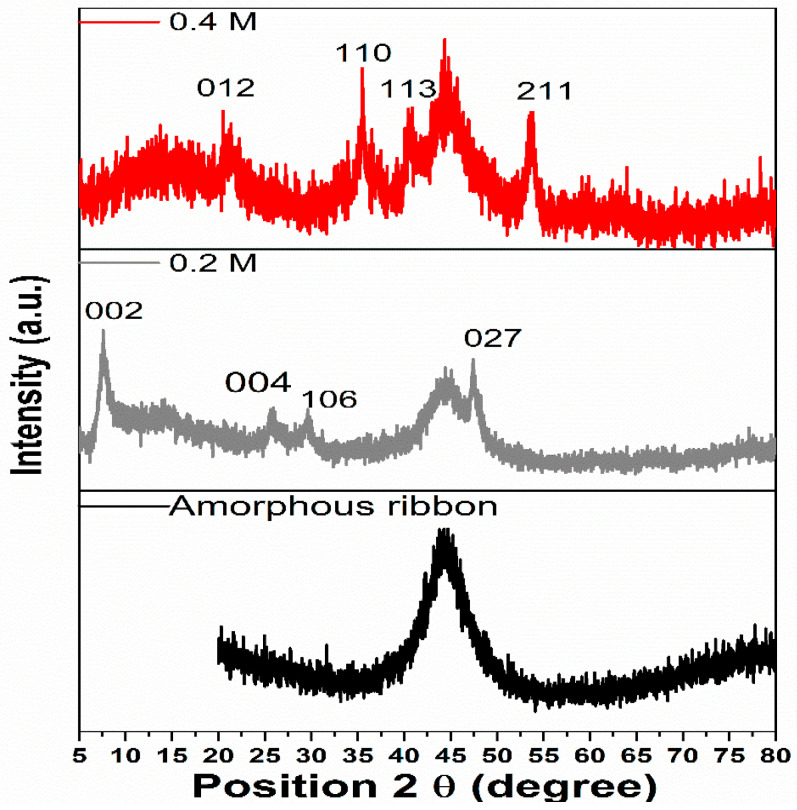
XRD pattern of amorphous ribbons and amorphous ribbons decorated with iron oxide nanoparticles.

**Figure 3 materials-16-02820-f003:**
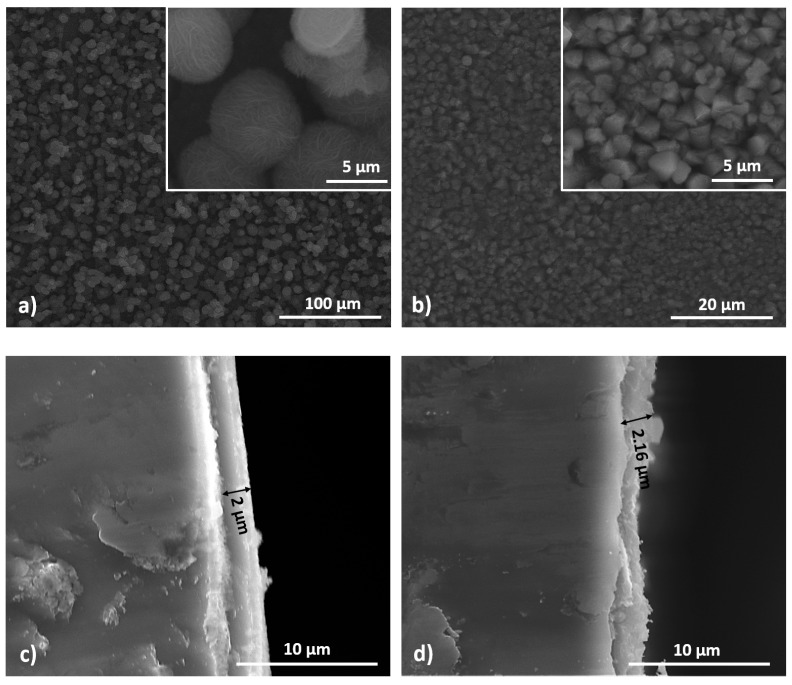
SEM images of the supercapacitor electrode surface obtained in a NaOH solution with a concentration of (**a**) 0.2 M and (**b**) 0.4 M. Cross-section images of electrodes obtained at (**c**) 0.2 M and (**d**) 0.4 M.

**Figure 4 materials-16-02820-f004:**
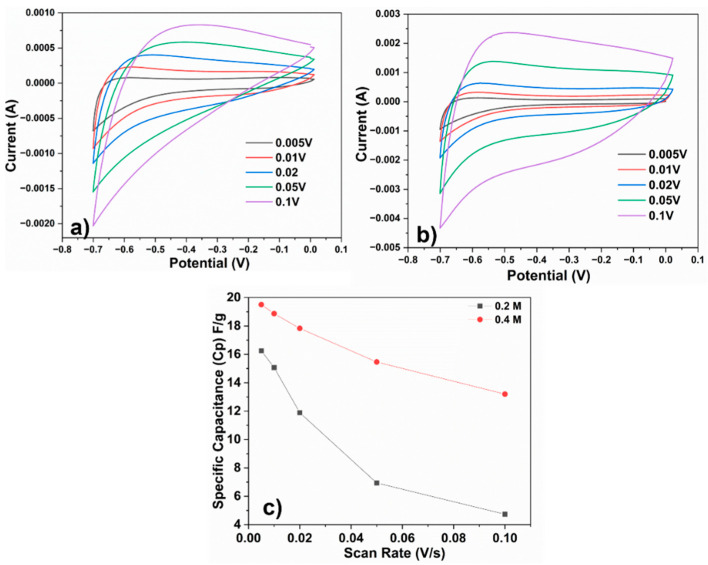
Cyclic voltammograms of (**a**) 0.2 M sample and (**b**) 0.4 M sample. (**c**) Plots of specific capacitance versus potential scan rate of samples.

**Figure 5 materials-16-02820-f005:**
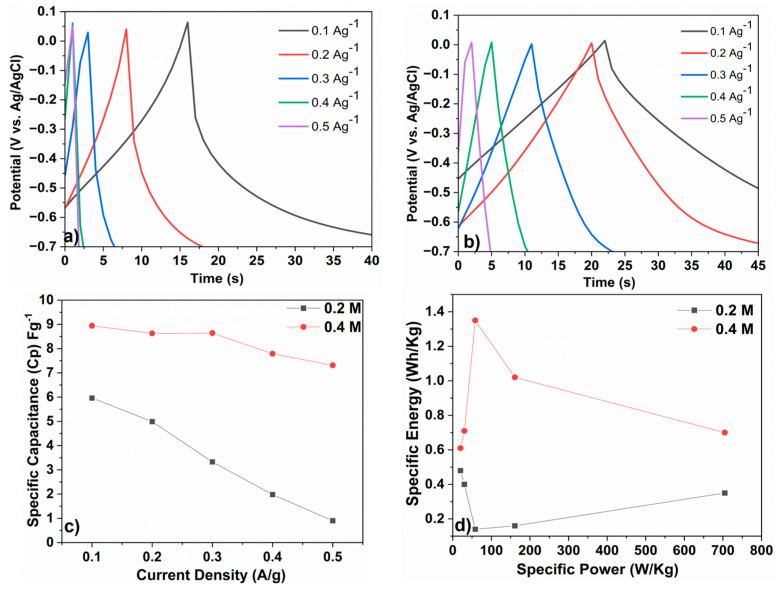
GCD curves of electrodes tested at different current densities for the (**a**) 0.2 M sample and (**b**) 0.4 M sample. (**c**) Specific capacitance of both electrodes at different current densities. (**d**) Ragone plot of both electrodes.

**Figure 6 materials-16-02820-f006:**
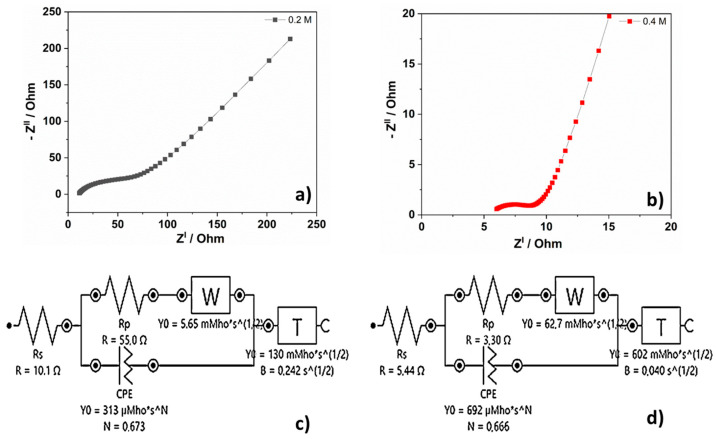
Nyquist plots of (**a**) 0.2 M sample and (**b**) 0.4 M sample. Equivalent circuit fitting of the samples: (**c**) 0.2 M and (**d**) 0.4 M.

## Data Availability

Not applicable.

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
