# Peer review of "Fabrication of Flexible Supercapacitor Electrode Materials by Chemical Oxidation of Iron-Based Amorphous Ribbons"

_materials, 2023, doi:10.3390/ma16072820_

Round 1
Reviewer 1 Report
The manuscript needs a major revision before it can be published in the journal. Some points are as follows:
-A comparing study on the chemical oxidation of iron-based ribbons should be conducted in the introduction. What is the novelty of this work, compared to other reports on this subject?
-What is the synthesis route for the preparation of the alloy? How about its characterization?
-Why did the authors choose the Fe75Si12B10Nb3 composition?
-How did the authors optimize conditions to get different crystal structures of iron oxide?
-What is the wavelength of the CuKa radiation? Is it CuKa or CuKa1?
-When XRD shows peaks then the material is not amorphous. It is a polycrystal or monocrystal. Please correct it in the text.
-What happens to other elements in the alloy? Why there is no evidence of their presence in the XRD pattern?
-SEM images are very low quality. Better images are needed. Also, a cross-section of the film is needed to measure the thickness of the oxide.
-A BET analysis is needed to talk about a specific area.
-The specific capacitance values should be compared with other reports.
Reviewer 2 Report
The manuscript entitled "Fabrication of flexible supercapacitor electrode materials by chemical oxidation of iron-based amorphous ribbons" reported the fabrication of flexible supercapacitors by chemical oxidation of iron based amorphous ribbons. The findings demonstrate promising potential for practical applications in energy storage devices. In general, it is an interesting and valuable topic to deserving a research article.
However, there are still many problems to be solved. So this reviewer would suggest a major revision before its acceptance.
1. One or two sentences to present the background or aim of this work should be added at the beginning of abstract.
2. Overall the draft is good but needs more careful editing.
3. When generally introduce the supercapacitors, some recent and important articles should be included: Journal of Bioresources and Bioproducts 7 (4), 245-269, 2022; Journal of Materials Chemistry A 8, 23059-23095, 2020; Journal of Materials Science 56, 173–200, 2021; etc.
4. More details on the raw materials should be provided to guarantee the work reproducible.
5. More experimental details should be added into figure 1.
6. To have a better readability and logic, it is better to divided the section 2 into several sub-sections including materials, methods, characterizations, etc.
7. It is better to provide SEM images with better readability.
8. The inset in Figure 6 should be modified to have a better readability.
9. To further show the novelty of this work, more comparison on the supercapacitance properties with previous work should be performed with supporting articles, such as: Sandwich-like chitosan porous carbon Spheres/MXene composite with high specific capacitance and rate performance for supercapacitors; Polymers 14 (13), 2521, 2022; New Journal of Chemistry 45 (48), 22602-22609, 2021; Chinese Chemical Letters 31 (7), 1986-1990, 2020; Polymer 235, 124276, 2021; Journal of Colloid and Interface Science 599, 443-452, 2021; Diamond and Related Materials 129, 109339, 2021; Diamond and Related Materials 130, 109526, 2022; Journal of Colloid and Interface Science 609, 179-187, 2022; https://doi.org/10.1007/s11705-022-2250-3; etc.
10. There are still some typos and grammar issues. In addition, please carefully check the references to ensure the full information is included.
